# Partial Substitution of Fishmeal with Mopane Worm Meal in Dusky Kob Fingerling (*Argyrosomus japonicus*) Diets: Feed Utilization, Digestive Enzyme Activity, Blood Parameters, and Growth Performance

Tshegofatso C. Nyuliwe [1], Victor Mlambo [1], Molatelo J. Madibana [2,*], Mulunda Mwanza [3] and Obiro C. Wokadala [1]

1   School of Agricultural Sciences, Faculty of Agriculture and Natural Sciences, University of Mpumalanga, Private Bag x11283, Mbombela 1200, South Africa; tshegofatso.nyuliwe@ump.ac.za (T.C.N.); victor.mlambo@ump.ac.za (V.M.); obiro.wokadala@ump.ac.za (O.C.W.)
2   Department of Forestry, Fisheries and the Environment, Martin Hammerschlag Way, Foreshore, Cape Town 8001, South Africa
3   Department of Animal Health, Faculty of Natural and Agricultural Sciences, North-West University, P Bag x2046, Mmabatho 2735, South Africa; mulunda.mwanza@nwu.ac.za
*   Correspondence: MMadibana@dffe.gov.za

**Abstract:** The use of protein-rich mopane worm meal (MPWM) in marine fish diets has the potential to reduce the overall demand for fishmeal (FM) and thus promote economically and ecologically sustainable dusky kob aquaculture. Therefore, this study investigated the effect of graded levels of MPWM on feed and nutrient utilization, digestive enzymes, blood parameters, and growth performance in dusky kob fingerlings (*Argyrosomus japonicus*, Temminck and Schlegel, 1843) over a 7-week feeding trial. Five isonitrogenous and isoenergetic commercial dusky kob diets were formulated by substituting 0 (MPWM0), 3 (MPWM3), 6 (MPWM6), 9 (MPWM9), and 18% (MPWM18) of FM with MPWM and offered at 2.8% of fish body weight. Two-thousand fingerlings (average weight $15.68 \pm 0.25$ g) were evenly distributed into 20 experimental tanks of a recirculating aquaculture system. Weight measurements were taken weekly, while blood and intestinal samples were taken at the end of the experiment. The inclusion of MPWM quadratically influenced ($y = 39.25 (\pm 1.27) + 0.89 (\pm 0.36)x - 0.04 (0.02) x^2$; $R^2 = 0.23$; $p = 0.03$) overall feed intake. Overall weight gain and specific growth rate (SGR) linearly increased while overall feed conversion ratio (FCR) linearly declined with MPWM inclusion levels. Substituting FM with MPWM did not alter ($p > 0.05$) digestive enzyme activities and haematological parameters. Among the serum biochemical components measured, only urea increased linearly in response to MPWM levels, while no trends were observed for the other components. It was concluded that replacing 11.13% of FM with MPWM in commercial dusky kob diet formulations does not compromise feed intake, feed utilization efficiency, growth rate, and physiological status of juvenile dusky kob.

**Keywords:** mopane worm meal; dusky kob; feed intake; growth; blood parameters; digestive enzymes

## 1. Introduction

Of the many challenges faced by the aquaculture industry, the ecological and economical unsustainability of fishmeal (FM) as an aquafeed ingredient is probably the most important. Consequently, there has been growing interest in the use of insect meals as FM alternatives in aquafeeds for various farmed fish over the last decade [1–3]. While the usage of insect meals in fish diets has been increasing in most parts of the world, sub-Saharan African countries have not yet realized the full potential of insects as protein sources in aquafeeds [3–5]). Of greater importance to the sustainable aquaculture drive are insects that

occur in large numbers in the wild and thus do not require high levels of care and inputs. One such insect that is commonly found in the wild in southern Africa is the mopane worm (*Imbrasia belina*) [4,5] which can reach up to 10 cm in length and has a distinct black colour with whitish-green and yellow bands. The worm is the larval stage of the emperor moth (Lepidoptera) [6] and mostly feeds on fresh leaves of the mopane tree (*Colophospermum mopane*), from which the insect derives its common name. It is commonly found in the semi-arid areas of South Africa, Namibia, Botswana, and Zimbabwe [7] and forms part of the diet for most rural communities in southern Africa. The worms are harvested in December and January as well as in April and May.

After harvesting, mopane worms are degutted by squeezing the intestinal contents out through their anal opening, cooked in brine and sun-dried [8]. The dried worm can be milled to produce mopane worm meal (MPWM), which has high protein (54–59%) and fat (12.9–16.7%) levels [8–10]. The MPWM is a source of highly digestible protein that has a good balance of essential and non-essential amino acids [11]. Because of these protein attributes, Rapatsa and Moyo [4] evaluated MPWM as a replacement of FM in diets of the omnivorous Mozambique tilapia (*Oreochromis mossambicus*). Higher thermal-unit growth coefficient (TGC), protein efficiency ratio (PER), and apparent digestibility coefficient (ADC) were observed with higher inclusion levels of MPWM in tilapia diets. However, when MPWM was evaluated in the omnivorous African catfish (*Clarius gariepinus*), Rapatsa and Moyo (2019) reported a decline in growth performance, PER, ADC, and the activity of some digestive enzymes such as amylase, while liver degradation scores increased with higher levels of MPWM. These contrasting findings on the utility of MPWM as an FM replacement suggest the need for further investigations using different types of fish. To our knowledge, no study has been undertaken to evaluate MPWM as a potential protein source in diets for a marine fish. Previously reported protein values of MPWM in omnivorous fish [4,5]) are unlikely to be the same as in the dusky kob, a carnivorous marine fish that is gaining popularity in South African aquaculture [3]. Therefore, the current study was designed to investigate the effect of graded levels (up to 18%) of MPWM on growth performance, intestinal enzyme activity, physiological parameters, and nutrient utilisation in dusky kob. The lower inclusion levels used in the current study (determined using a computer-assisted diet formulation) were informed by contrasting results, achieved when higher levels (up to 60%) were used in both tilapia [4] and catfish diets [5].

## 2. Materials and Methods

### 2.1. Ethical Statement

This seven-week feeding trial was commissioned and conducted by adhering to protocols highlighted in the South African Animals Protection Act, 1962 (Act 71 of 1962). The study was approved by the Animal Research Ethics Committee of the University of Mpumalanga (FANS17) and the Aquaculture Animal Ethics Committee (AAEC) at the Fisheries Branch of the Department of Forestry, Fisheries and Environment (DFFE) (20201111_dk_01_Mlambo).

### 2.2. Mopane Worms

The worms were procured from Tsetsebjwe village, Botswana (22.415313° S 28.394753° E). The worms were degutted and the gut contents were squeezed out of the body by pushing the head towards the anal opening as described by Siame et al. [12]. Following the removal of the gut contents, the worms were washed and then cooked in brine. Immediately after cooking, the worms were sun-dried until they were moisture free (constant weight), before being packaged in woven polypropylene bags. Prior to their inclusion in the experimental diets, dried worms were ground using a laboratory scale rotor beater mill (SR 300, Retsch, Germany) fitted with a 500-micron mesh to produce mopane worm meal (MPWM).

## 2.3. Experimental Diets

Five isonitrogenous and isoenergetic dietary treatments were formulated by partially replacing FM with MPWM in a commercial dusky kob diet formulation (SA Feed (Pty) Ltd., Hermanus, South Africa) as follows: 1. commercial dusky kob diet in which FM was not replaced with MPWM (MPWM0); 2. commercial dusky kob diet in which 3% of FM was replaced with MPWM (MPWM3); 3. commercial dusky kob diet in which 6% of FM was replaced with MPWM (MPWM6); 4. commercial dusky kob diet in which 9% of FM was replaced with MPWM (MPWM9); 5. commercial dusky kob diet in which 18% of FM was replaced with MPWM (MPWM18). The ingredients were hand mixed at the Marine Research Aquarium (MRA) of the Department of Forestry, Fisheries and the Environment in Cape Town, South Africa. Water was added to the ingredient mixture and carefully kneaded to produce a dough that was then pasted on a plastic sheet as a thin layer. The layer was later dried to constant mass using a household floor fan. A maize kernel hand grinder with an adjustable pressure disc was used to produce 2 mm flakes.

## 2.4. Analyses of Experimental Diets and Mopane Worm Meal

Diets and mopane worm meal were sampled and analysed for dry matter (DM), ash, crude fat, and crude protein content according to standard AOAC [13] methods. AOAC method numbers 930.15, 942.05, 945.16, and 976.05 were used for the analysis of DM, ash, crude fat, and total nitrogen, respectively. Total nitrogen (N) was converted to crude protein (CP) by multiplying it by a factor of 6.25. To determine digestible energy (DE), dietary gross energy (GE) was first determined by using an adiabatic oxygen bomb calorimeter (Parr Instruments, Moline, IL, USA) with benzoic acid as a standard. The reported DE is the difference between faecal GE and dietary GE. Metabolizable energy (ME) was calculated by subtracting the amount of energy lost through the nitrogenous excretion (NE) from DE. The energy loss through NE was assumed to be 11% of dietary GE. Amino acids (AA) analysis was performed using HPLC on acid-hydrolysed samples according to the method described by Einarsson et al. [14]. Detection of AA was done with the aid of a fluorescence detector, and the concentration was expressed in g/100 g of sample. Mineral content was determined using an ICP Mass Spectrometer (Perkin-Elmer, 1982, NexION 300Q). Chitin extraction and characterization was done using C solid-state NMR and FT-IR spectroscopy and electron microscopy, according to the methods described by Zhang et al. [15] and Majtan et al. [16].

## 2.5. Feeding Trial

An automated recirculating aquaculture system (RAS) at MRA, comprising 20 high-density polyethylene tanks (465 L, 67 cm deep, and 94 cm in diameter) coated with white fiberglass resin to increase fish visibility, was used. The RAS was connected to the supporting water filtration components, which included a sand filter, biological filtration media tank, and foam fractionator. The desired sea water temperature of 25 °C was maintained through a heat pump (AquaHeat, Cape Town, South Africa). Air (5.5–6 mg/L dissolved oxygen) was brought into the RAS by air lines connected to an air blower. The natural seawater salinity was 34 ppt. Water temperature, dissolved oxygen, and salinity were monitored using a YSI multiparameter instrument (YSI Incorporated, Yellow Springs, OH, USA). Ammonia concentration was monitored twice weekly using a Sera ammonium/ammonia test kit (North Rhine-Westphalia, Germany), and the concentration remained below 0.05 mg/L throughout the 7-week feeding trial.

A total of 2000 dusky kob fingerlings with initial average weight of 15.68 ± 0.25 g at the start of the experiment were purchased from Kingfish Enterprises PTY LTD, East London, South Africa, and transported to MRA as previously described in detail by Mdhluvu et al. [17]. Fish were acclimatized to the experimental system and the diets for three weeks prior to the commencement of the feeding trial. Each experimental diet was randomly allocated to four replicate tanks (experimental units) carrying 100 fingerlings each. To minimise cannibalism, which is commonly observed in dusky kob fingerlings, no artificial

lighting was provided. For the entire seven weeks of the feeding trial, fish were offered diets by hand in the morning (08:00) and in the afternoon (15:00) at an optimum feeding intensity of 2.8% of their body weight, as previously determined by Madibana & Mlambo [18]. At this feeding intensity, juvenile dusky kob do not demonstrate any feed refusals. In each of the twenty tanks, twenty fish (80 per replicate) were subjected to weekly measurements of body mass to determine the weight gain, specific growth rate (SGR), and feed conversion ratio (FCR). Feed intake (FI) and protein intake were also determined. The amount of feed offered in a particular week was calculated as 2.8% of the fish body weight of the preceding week. No mortalities were recorded after the commencement of the feeding trial.

*2.6. Calculations*

Since feeding intensity was set at a level that did not leave any refusals, the amount of feed offered was also the amount consumed. Protein intake was determined by multiplying the amount of feed consumed by its protein content, as analysed in the laboratory. Specific growth rate (SGR, % body weight gain/day) was calculated based on the formula below:

$$\text{SGR} = \frac{\text{In}(\text{final weight}) - \text{In}(\text{Initial weight})}{\text{Time intervals (days)}} \times 100$$

Feed conversion ratio (FCR) was calculated as follows:

$$\text{FCR} = \frac{\text{Feed consumed (g)}}{\text{Fish weight gain (g)}}$$

Protein efficiency ratio (PER) was calculated as follows:

$$\text{PER} = \frac{\text{Weight gain (g)}}{\text{Protein intake (g)}}$$

*2.7. Haematology and Serum Biochemical Analyses*

At trial termination, six randomly selected fish from each tank were anaesthetized, using 2-phenoxyethanol (Sigma-Aldrich, St. Louis, MO, USA), for blood collection. Under anaesthesia, the fish caudal fin was ablated and bled into two different types of tubes. Tubes containing EDTA anticoagulant were used for blood meant for haematological analyses. Light microscope under oil immersion at ×100 magnification (Olympus, Tokyo, Japan) was used for manual blood count of thrombocytes, lymphocytes, monocytes, neutrophils, basophils, and eosinophils. Freshly sampled blood was processed using microhematocrit centrifuge (NI 1807 Nova Instruments, Berkshire, UK) for 5 min at 10,000 rpm (12,298 g) to determine haematocrit values. Blood samples left to clot in EDTA-free tubes at room temperature were centrifuged for 15 min at 3500 rpm to harvest serum for determination of serum biochemical parameters. The serum was transferred to sterilized bottles for further analysis. The modified method of Wright et al. [19] was performed to measure alkaline phosphatase (ALP) activity. Aspartate aminotransferase (AST) and alanine transaminase (ALT) activities were determined based on the methods developed by Reitman and Frankel [20], while the Biuret method standardized for the RA-1000 (Technicon method no. SM4-0147K82, 1982) was performed for total protein (TP) determination. Technicon method numbers SM4-0131K82 (1982) and SM4-0141K82 (1982) were used to determine albumin and creatinine, respectively. Total globulin fraction was determined by subtracting the albumin from the TP. A standard RA-1000 enzymatic method was employed for the analysis of triglycerides using the Boehringer Mannheim GPO-PAP kit, while the monocholesterol (CHOD-PAP) method was used to determine cholesterol content. Urea content was estimated according to the Crest Biosystems Modified Berthelot method [21].

*2.8. Enzyme Activity*

Ten fish from each tank were anaesthetised (2-phenoxyethanol) before the entire intestinal section from each fish was cut and preserved in 10% formalin, pending further processing for enzymatic analyses. Preparation of the intestinal samples for enzymatic activity analyses was conducted as described by Rapatsa and Moyo [5]. For intestinal amylase activity, starch (1% $w/v$) was prepared and quantified as described by Bernfeld [22]. Lipase activity was determined following the method described by Markweg et al. [23]. Protease activity was determined using 1% ($w/v$) azocasein, as described by Bezerra et al. [24]. For chitinase activity determination, chitin-azure was used as a substrate, as described by Hood and Meyers [25] and modified by Rapatsa and Moyo [5]. The enzyme analyses were performed at Sonto-Rose Solutions (Pty) Ltd. in Polokwane, South Africa.

*2.9. Statistical Analysis*

Measurements from multiple fish per tank were averaged before analysis, such that each replicate tank had one value for each parameter of interest. Data were also analysed for homogeneity of variance and for normality using Levene's test and the NORMAL option in the Procedure Univariate statement, respectively. Dietary effects on overall feed utilization, growth performance, enzyme activity, and blood parameters were analysed using the general linear model (GLM) procedure of SAS [26]. Weekly measured parameters (feed intake, protein intake, weight gain, specific growth rate, feed conversion ratio) were analysed using the repeated measures analysis [26]. Data from all measured parameters were evaluated for linear and quadratic effects using response surface regression analysis (RSREG) [26] to describe the responses of dusky kob to incremental levels of mopane worm according to the following quadratic model: $y = c + bx + ax^2$, where $y$ = response variable, $a$ and $b$ are the coefficients of the quadratic equation, $c$ is intercept, $x$ is dietary mopane worm level (%), and $-b/2a$ is the mopane worm value at the maximum or minimum point of the quadratic response. Least-square means were compared using the probability of difference option of the lsmeans statement [26], and the significance level was set at $p \leq 0.05$.

**3. Results**

*3.1. Dietary Composition*

Table 1 shows that all the dietary treatments, except MPWM3, had similar levels of ash, dry matter, and moisture. The crude fat and crude protein contents were similar for all dietary treatments. The energy content was also similar across the treatments. Mopane worm meal inclusion diets had numerically higher leucine, lysine, serine, aspartic acid, and tyrosine levels compared to the control diet. The energy content of the five diets was statistically similar.

*3.2. Feed Utilization and Growth Performance*

Diet × week interaction effect was significant ($p < 0.05$) for weight gain (Table 2) but not ($p > 0.05$) for specific growth rate (SGR), feed intake, FCR, protein intake, and protein efficiency ratio (PER) (Table 3). After the first week of feeding, the groups fed MPWM6, MPWM9, and MPWM18 exhibited closely similar numerical weight gain, with MPWM0 ranking the lowest in terms of weight gain. In the second week, the MPWM3-fed group showed numerically higher weight gain, followed by the MPWM18-fed group, with MPWM9 ranking lowest. The third week saw all the groups fed MPWM inclusion diets showing closely similar numerical weight gain, with the control-fed group exhibiting the lowest numerical weight gain. The group fed MPWM6 had numerically superior weight gain to the other groups in both week 4 and week 5 of feeding (Table 2), with MPWM3 and MPWM9 ranking the lowest in the respective weeks. In week 6, the MPWM3-fed group had numerically higher weight gain compared to the rest, with MPWM0 ranking the lowest. The MPWM18-fed group showed a significantly higher weight gain compared to the rest of the groups in the final week, with MPWM6 showing the lowest weight gain in the final week. There were neither linear nor quadratic dietary effects on fish weight gain from week

1 to week 3 and week 5 and week 6. There was, however, a quadratic trend observed in weeks 4 ($y$ = 4.48 ($\pm$1.01) + 0.89 ($\pm$0.29)$x$ − 0.03 (0.01)$x^2$; $R^2$ = 0.22; $p$ = 0.04) and a linear trend observed in week 7 ($y$ = 6.30 ($\pm$1.31) − 0.23 ($\pm$0.37)$x$; $R^2$ = 0.38; $p$ = 0.00). A linear trend was also observed on overall weight gain ($y$ = 35.63 (2.53) + 1.05 (0.72)$x$; $R^2$ = 0.26; $p$ = 0.02) as the level of dietary MPWM increased.

**Table 1.** Experimental diet formulae and chemical composition of the diets and mopane worm meal (MPWM).

| | Diets [1] | | | | | MPWM |
|---|---|---|---|---|---|---|
| | MPWM0 | MPWM3 | MPWM6 | MPWM9 | MPWM18 | |
| **Ingredients (%)** | | | | | | |
| Mopane worm meal | 0.00 | 3.00 | 5.98 | 9.00 | 18.00 | |
| Fishmeal 66 | 30.00 | 26.95 | 23.95 | 21.00 | 12.00 | |
| Maize | 18.71 | 18.27 | 17.84 | 17.46 | 16.21 | |
| Soya oilcake | 10.00 | 9.15 | 8.32 | 7.50 | 5.00 | |
| Full fat soya 58 | 8.50 | 7.92 | 7.35 | 6.80 | 5.11 | |
| Blood meal 90 | 6.00 | 5.74 | 5.49 | 5.25 | 4.50 | |
| Wheat gluten | 10.91 | 11.71 | 12.52 | 13.38 | 15.85 | |
| Pork meal 28 | 8.17 | 9.34 | 10.51 | 11.72 | 15.27 | |
| Fish and poultry oil | 5.36 | 5.47 | 5.58 | 5.72 | 6.07 | |
| Vit/Min Premix | 2.35 | 2.44 | 2.47 | 2.13 | 0.8 | |
| **Proximate composition (calculated values except for MPWM)** | | | | | | |
| Dry matter (%) | 86.80 | 90.50 | 87.80 | 88.40 | 86.50 | 87.84 |
| Ash (% DM) | 0.62 | 1.34 | 0.48 | 0.39 | 0.79 | 12.16 |
| Moisture (% DM) | 13.20 | 9.50 | 12.20 | 11.60 | 13.55 | 7.04 |
| Crude protein (% DM) | 46.58 | 46.57 | 46.56 | 46.55 | 46.51 | 55.71 |
| Crude fat (% DM) | 12.15 | 12.15 | 12.15 | 12.14 | 12.13 | 6.52 |
| Digestible energy (MJ/kg DM) | 17.38 | 17.34 | 17.38 | 17.37 | 17.36 | - |
| Metabolisable energy (MJ/kg DM) | 15.75 | 15.66 | 15.57 | 15.47 | 15.20 | - |
| Chitin (% DM) | 0.50 | 0.50 | 0.70 | 1.3 | 1.9 | 13.52 |
| **Essential amino acids (g/100 g DM)** | | | | | | |
| Arginine | 3.54 | 4.62 | 4.75 | 5.02 | 5.21 | 6.32 |
| Threonine | 1.66 | 2.27 | 2.12 | 2.33 | 2.38 | 2.37 |
| Methionine | 0.71 | 1.33 | 1.22 | 1.42 | 1.35 | 0.68 |
| Valine | 2.47 | 3.04 | 2.12 | 3.09 | 3.25 | 2.88 |
| Phenylalanine | 1.93 | 2.17 | 2.22 | 2.19 | 2.30 | 2.17 |
| Isoleucine | 1.98 | 2.90 | 3.02 | 2.98 | 3.05 | 2.06 |
| Leucine | 3.42 | 4.03 | 4.11 | 4.05 | 4.03 | 3.03 |
| Histidine | 1.34 | 2.24 | 2.33 | 2.44 | 1.11 | 2.69 |
| Lysine | 2.92 | 5.40 | 5.55 | 5.58 | 5.91 | 3.41 |
| Tryptophan | 5.40 | 5.99 | 6.11 | 6.12 | 6.10 | - |
| **Non-essential amino acids (g/100 g DM)** | | | | | | |
| Alanine | 2.45 | 2.63 | 2.77 | 2.88 | 2.99 | 2.53 |
| Tyrosine | 1.22 | 2.17 | 2.22 | 2.11 | 1.98 | 2.17 |
| Proline | 2.45 | 2.25 | 2.32 | 2.55 | 2.68 | 2.55 |
| Serine | 1.85 | 2.12 | 2.13 | 2.23 | 2.24 | 2.49 |
| Aspartic acid | 3.36 | 4.57 | 4.65 | 4.66 | 4.81 | 4.46 |
| Glutamic acid | 6.17 | 7.65 | 7.52 | 7.72 | 7.91 | 6.16 |
| Glycine | 2.68 | 2.98 | 2.88 | 3.21 | 4.18 | 2.65 |

[1] Diets: Formulated by substituting 0 (MPWM0), 3 (MPWM3), 6 (MPWM6), 9 (MPWM9), and 18% (MPWM18) of fishmeal with mopane worm meal. Mopane worm meal was supplied from an informal market in Tsetsebjwe village, Botswana. Fishmeal, wheat gluten, pork meal, maize, fish and poultry oil, and vit/min premix were all sourced from Johannesburg, South Africa. Blood meal was sourced from Germany. Soya oilcake and full-fat soya were sourced from Argentina. Poultry meal was sourced from Netherlands.

Increasing the level of MPWM in juvenile dusky kob had a quadratic effect on overall feed intake ($y$ = 39.25 ($\pm$1.27) + 0.89 ($\pm$0.36)$x$ − 0.04 (0.02)$x^2$; $R^2$ = 0.23; $p$ = 0.03). The diet with the highest MPWM (MPWM18) promoted the lowest overall feed conversion ratio (FCR) value (0.91), while the control diet recorded the highest FCR value of 1.13. There

was a significant negative linear trend for overall FCR ($y = 1.11 \ (\pm0.05) - 10.01 \ (\pm0.01)x$; $R^2 = 0.30$; $p = 0.01$) as MPWM levels increased in juvenile dusky kob diets. A quadratic dietary effect was observed on overall fish protein intake ($y = 18.25 \ (\pm0.59) + 0.41 \ (\pm0.16)x - 0.02 \ (0.01)x^2$; $R^2 = 0.23$; $p = 0.03$) as the levels of dietary MPWM increased. Overall protein efficiency ratio (PER) marginally increased with an increase of dietary MPWM ($p > 0.05$), from 1.91 for MPWM0-fed fish to 2.35 for MPWM18-fed fish (Table 3). There was a significant linear increase in PER ($y = 1.94 \ (\pm1.10) + 0.01 \ (\pm0.03)x$; $R^2 = 0.33$; $p = 0.01$) as the levels of dietary MPWM increased. Fish specific growth rate (SGR) increased linearly ($y = 2.41 \ (\pm0.10) + 0.03 \ (\pm0.02)x$; $R^2 = 0.32$; $p = 0.01$) in response to incremental levels of dietary MPWM.

**Table 2.** Average weekly fish weight gain (g) of dusky kob, *Argyrosomus japonicus*, in response to increasing levels of dietary mopane worm meal (MPWM).

| | Diets [1] | | | | | SEM [2] | Significance [3] | |
|---|---|---|---|---|---|---|---|---|
| | MPWM0 | MPWM3 | MPWM6 | MPWM9 | MPWM18 | | Linear | Quadratic |
| Week 1 | 2.72 | 3.50 | 3.77 | 3.81 | 3.79 | 0.63 | NS | NS |
| Week 2 | 4.71 | 6.07 | 4.76 | 4.50 | 5.72 | 1.02 | NS | NS |
| Week 3 | 3.71 | 5.52 | 5.96 | 5.68 | 5.31 | 1.20 | NS | NS |
| Week 4 | 5.04 | 4.34 | 8.36 | 6.73 | 4.64 | 1.03 | NS | * |
| Week 5 | 5.58 | 7.83 | 7.90 | 5.12 | 7.76 | 1.34 | NS | NS |
| Week 6 | 5.11 | 8.61 | 6.93 | 5.66 | 5.93 | 1.60 | NS | NS |
| Week 7 | 6.98 [a,b] | 5.49 [a] | 4.08 [a] | 9.87 [a,b] | 12.31 [b] | 1.34 | * | NS |
| Overall gain | 33.85 | 41.36 | 41.76 | 40.37 | 45.48 | 2.79 | * | NS |

[1] Diets: Formulated by substituting 0 (MPWM0), 3 (MPWM3), 6 (MPWM6), 9 (MPWM9), and 18% (MPWM18) of fishmeal with mopane worm meal [2] SEM: standard error of the mean. [3] Significance: NS = $p > 0.05$; * = $p < 0.05$. [a,b] Means in the same row with common superscripts do not differ ($p > 0.05$).

**Table 3.** Overall feed intake (g/fish/day), protein intake (g/fish), specific growth rate (SGR), feed conversion ratio (FCR), and protein efficiency ratio (PER) of dusky kob, *Argyrosomus japonicus*, in response to increasing levels of mopane worm meal.

| | Diets [1] | | | | | SEM [2] | Significance [3] | |
|---|---|---|---|---|---|---|---|---|
| | MPWM0 | MPWM3 | MPWM6 | MPWM9 | MPWM18 | | Linear | Quadratic |
| Feed intake | 38.14 [a] | 43.10 [a,b] | 44.11 [b] | 41.92 [a,b] | 41.61 [a,b] | 0.70 | NS | * |
| Protein intake | 17.73 [a] | 20.04 [a,b] | 20.50 [b] | 19.49 [a,b] | 19.35 [a,b] | 0.32 | NS | * |
| SGR (% per day) | 2.36 | 2.58 | 2.60 | 2.60 | 2.86 | 0.06 | * | NS |
| FCR | 1.13 | 1.06 | 1.06 | 1.06 | 0.91 | 0.03 | * | NS |
| PER | 1.91 | 2.05 | 2.04 | 2.07 | 2.35 | 0.06 | * | NS |

[1] Diets: Formulated by substituting 0 (MPWM0), 3 (MPWM3), 6 (MPWM6), 9 (MPWM9), and 18% (MPWM18) of fishmeal with mopane worm meal [2] SEM: standard error of the mean. [3] Significance: NS = $p > 0.05$; * = $p < 0.05$. [a,b] Means in the same row with common superscripts do not differ ($p > 0.05$).

### 3.3. Haematology and Serum Biochemistry

Dietary treatments had a significant effect on monocyte and basophil levels ($p < 0.05$) but not on haematocrit, thrombocytes, eosinophils, neutrophils, and lymphocytes ($p > 0.05$). The control diet promoted the highest levels of basophils compared to the other four diets with mopane worm meal (Table 4). There were neither linear nor quadratic dietary effects on all the haematology parameters ($p > 0.05$). The dietary treatments had a significant effect on urea, total protein, albumin, and triglyceride concentration ($p < 0.05$) but not on creatinine, globulin, albumin:globulin, alanine aminotransferase, aspartate aminotransferase, alkaline phosphatase, and cholesterol concentrations ($p > 0.05$) (Table 5). Serum triglyceride levels linearly declined ($y = 2.88 \ (\pm0.53) - 0.10 \ (\pm0.15)x$; $R^2 = 0.27$; $p = 0.02$), while urea linearly increased as MPWM inclusion levels increased ($y = 1.98 \ (\pm0.23) + 0.08 \ (\pm0.07)x$; $R^2 = 0.37$; $p = 0.01$).

**Table 4.** Effect of graded levels of mopane worm meal on some haematological parameters of juvenile dusky kob, *Argyrosomus japonicus*.

| | Diets [1] | | | | | SEM [2] | Significance [3] | |
|---|---|---|---|---|---|---|---|---|
| | MPWM0 | MPWM3 | MPWM6 | MPWM9 | MPWM18 | | Linear | Quadratic |
| Haematocrit (%) | 31.62 | 32.10 | 36.62 | 33.50 | 34.12 | 0.92 | NS | NS |
| Thrombocytes ($mm^3$) | 5.23 | 3.13 | 4.13 | 2.85 | 5.05 | 0.52 | NS | NS |
| Lymphocytes (%) | 88.50 | 89.25 | 94.00 | 80.25 | 83.50 | 1.68 | NS | NS |
| Monocytes (%) | 2.75 [a] | 5.50 [a,b] | 2.00 [a] | 9.25 [b] | 6.25 [a,b] | 0.80 | NS | NS |
| Neutrophils (%) | 0.00 | 0.00 | 0.50 | 1.50 | 0.25 | 0.22 | NS | NS |
| Basophils (%) | 6.00 [b] | 3.00 [a,b] | 1.75 [a] | 4.24 [a,b] | 3.00 [a,b] | 0.45 | NS | NS |
| Eosinophils ($10^3$ cells $Ml^{-1}$) | 2.25 | 2.25 | 1.75 | 4.75 | 6.75 | 0.93 | NS | NS |

[1] Diets: Formulated by substituting 0 (MPWM0), 3 (MPWM3), 6 (MPWM6), 9 (MPWM9), and 18% (MPWM18) of fishmeal with mopane worm meal. [2] SEM: standard error of the mean. [3] Significance: NS = $p > 0.05$; * = $p < 0.05$. [a,b] Means in the same row with common superscripts do not differ ($p > 0.05$).

**Table 5.** Effect of graded levels of mopane worm meal on concentration of selected blood serum metabolites in dusky kob, *Argyrosomus japonicus*.

| | Diets [1] | | | | | SEM [2] | Significance [3] | |
|---|---|---|---|---|---|---|---|---|
| | MPWM0 | MPWM3 | MPWM6 | MPWM9 | MPWM18 | | Linear | Quadratic |
| Urea ($mmol\ L^{-1}$) | 2.23 [a,b] | 1.85 [a] | 2.28 [a,b] | 2.97 [b] | 3.03 [b] | 0.14 | * | NS |
| Creatinine ($mmol\ L^{-1}$) | 9.00 | 9.00 | 22.00 | 9.00 | 9.00 | 1.79 | NS | NS |
| Total Protein ($mmol\ L^{-1}$) | 40.00 [a] | 40.75 [a,b] | 45.75 [b] | 42.50 [a,b] | 43.25 [a,b] | 0.69 | NS | NS |
| Albumin ($g\ L^{-1}$) | 14.00 [a,b] | 13.00 [a] | 12.25 [b] | 14.25 [a,b] | 14.50 [b] | 0.21 | NS | NS |
| Globulin ($g\ L^{-1}$) | 26.00 [a] | 27.75 [a,b] | 30.50 [b] | 28.28 [a,b] | 28.75 [a,b] | 0.50 | NS | NS |
| Albumin:Globulin | 0.53 | 0.50 | 0.50 | 0.50 | 0.50 | 0.01 | NS | NS |
| Alanine aminotransferase ($U\ L^{-1}$) | 24.25 | 25.00 | 37.75 | 29.25 | 28.50 | 3.92 | NS | NS |
| Aspartate aminotransferase ($U\ L^{-1}$) | 82.50 | 87.50 | 114.00 | 102.00 | 92.75 | 12.91 | NS | NS |
| Alkaline phosphatase ($U\ L^{-1}$) | 30.50 | 29.00 | 35.50 | 35.00 | 38.00 | 1.62 | NS | NS |
| Cholesterol ($mmol\ L^{-1}$) | 1.77 | 1.79 | 1.94 | 1.80 | 1.97 | 0.06 | NS | NS |
| Triglycerides ($mmol\ L^{-1}$) | 2.92 | 2.32 | 3.47 | 2.45 | 4.86 | 0.31 | * | NS |

[1] Diets: Formulated by substituting 0 (MPWM0), 3 (MPWM3), 6 (MPWM6), 9 (MPWM9), and 18% (MPWM18) of fishmeal with mopane worm meal. [2] SEM: standard error of the mean. [3] Significance: NS = $p > 0.05$; * = $p < 0.05$. [a,b] Means in the same row with common superscripts do not differ ($p > 0.05$).

### 3.4. Digestive Enzymes

Experimental diets had no significant effect on the activity of all intestinal enzymes measured ($p > 0.05$). Table 6 shows that the group fed MPWM3 diets had numerically superior intestinal amylase, lipase, and protease activity. All dietary treatments exhibited numerically similar chitinase activity in response to MPWM inclusion in the diets. There were neither linear nor quadratic trends in the activity of all the enzymes in response to increasing MPWM levels in diets.

**Table 6.** Dusky kob intestinal amylase, lipase, protease, and chitinase activity (μmol/mg/min) in response to increasing levels of dietary mopane worm meal (MPWM).

| | [1] Diets | | | | | |
|---|---|---|---|---|---|---|
| Enzymes | MPWM0 | MPWM3 | MPWM6 | MPWM9 | MPWM18 | [2] Significance |
| Amylase | 496.61 ± 153.46 | 1017.18 ± 195.09 | 513.63 ± 166.51 | 616.49 ± 149.70 | 475.61 ± 151.82 | NS |
| Lipase | 11.16 ± 1.37 | 16.39 ± 1.56 | 14.39 ± 1.50 | 12.482 ± 1.47 | 12.11 ± 1.95 | NS |
| Protease | 273.53 ± 15.51 | 285.01 ± 55.39 | 245.24 ± 10.44 | 266.50 ± 4.48 | 229.52 ± 14.81 | NS |
| Chitinase | 0.02 ± 0.00 | 0.03 ± 0.00 | 0.03 ± 0.00 | 0.03 ± 0.00 | 0.03 ± 0.00 | NS |

[1] Diets: Formulated by substituting 0 (MPWM0), 3 (MPWM3), 6 (MPWM6), 9 (MPWM9), and 18% (MPWM18) of fishmeal with mopane worm meal. [2] Significance: NS = $p > 0.05$.

## 4. Discussion

### 4.1. Feed Utilization, Growth Performance, and Enzyme Activity

Previously, the utility of MPWM as an aquafeed ingredient was evaluated in two freshwater fish species, tilapia (*Oreochromis mossambicus*) [4] and African catfish (*Clarias gariepinus*) [5] but not in any marine fish species. Therefore, the current study is the first report on the dietary effects of MPWM in the dusky kob, a marine fish species. A quadratic effect on feed intake in response to increasing levels of MPWM in the diets suggests that there was a decline in feed intake at higher MPWM inclusion levels. Based on this quadratic trend, the maximum tolerance level of MPWM by juvenile dusky kob was calculated to be 11.13%. The only study to evaluate incremental levels of an insect meal (black soldier fly larvae meal) as an FM alternative in juvenile dusky kob diets [3] revealed neither a linear nor quadratic effect on fish feed intake. The reduction in feed intake at higher MPWM inclusion levels could be attributed to the presence of chitin in the insect meal. Indeed, Gopalakannan and Arul [27] and Kroeckel et al. [28] reported that diets with as low as 1% chitin may reduce feed intake in fish. The level of chitin in the MPWM used in the current study was 13.5%, which resulted in chitin levels of 1.3 and 1.9% in MPWM9 and MPWM18, respectively. This could be the reason why feed intake started to decline once inclusion levels reached 9% (MPWM9) and above. Kroeckel et al. [28] suggested that higher dietary inclusion levels of insect meals (>33%) are likely to reduce palatability, feed intake, and protein intake, resulting in the poor growth performance of fish. It has also been reported that substantial substitution of FM has the potential to reduce palatability of diets in some fish species [29]. These findings contradict Rapatsa and Moyo [5], who reported that feed intake in *Clarias gariepinus* was not affected by increasing levels of MPWM. This discordance can be attributed to the different fish species (marine carnivorous dusky kob and freshwater omnivorous African catfish) used in the two studies. In addition, chitinase activity in juvenile dusky kob was very low in the current study, whereas Rapatsa and Moyo [5] recorded 50 times more chitinase activity in the intestine of African catfish, thus explaining why no reduction of feed intake was observed with MPWM inclusion in catfish. The feed conversion ratio in fish fed MPWM-containing diets was lower (better) than that of the control diet, a possible reflection of higher growth performance in MPWM-containing diets. Specific growth rate in the current study increased with dietary MPWM inclusion levels in contrast to the findings reported by Rapatsa and Moyo [5] in C. *gariepinus*. These authors reported the best growth performance in the control group, whereas in the current study, the best growth rate was in the MPWM18 group, which received the highest MPWM. Danulat [30] reported that chitin-rich diets increase digestive enzyme activities in some fish, thereby accelerating nutrient utilisation for growth purposes. At 10% chitin inclusion levels, growth was significantly enhanced in sea bream (*Pagrus major*), Japanese eel (*Aguilla japonica*), and yellowtail (*Seriola quinqueradiata*) [31]. Some studies conducted over 50 years ago also shed light on how fish species devoid of chitinase can process chitin-containing diets with no adverse effect on growth or FCR. In their seminal work, Sera and Kimata [32] noticed that the population of allochthonous heterotrophic and chitin-decomposing bacteria in the stomach and intestines of sea bream (*Evynmis japonica*) was higher in fish fed the chitin-supplemented diets. Some reports also suggest that chitin has immune system–boosting properties [33,34], which affords animals the opportunity to channel the energy required to utilise nutrients derived from their diets rather than combating gut or external pathogens.

Protein efficiency ratio tended to increase as dietary MPWM inclusion increased, in contrast to Rapatsa and Moyo [5], who reported a decrease in PER in response to incremental levels of MPWM in *C. gariepinus* diets. The difference in PER response may be due to differences in the inclusion levels of MPWM. Rapatsa and Moyo [5] included up to 60% of chitin-containing MPWM in catfish diets, which may have decreased PER as MPWM increased. Additionally, high dietary protein levels often result in the catabolism of amino acids, which is associated with excretion and loss of energy [35]. Catfish require around 40% protein [36] in their diets for optimum PER. Kroeckel et al. [28] also reported

a significant decrease in PER in juvenile turbot (*Psetta maxima*) when fed diets containing black soldier fly meal.

There was no dietary effect on the activity of all the digestive enzymes measured in the current study. This suggests that inclusion of MPWM in juvenile dusky kob diets did not induce changes in digestive enzyme secretion. Insects, unlike plant protein sources [37], are not associated with universally known antinutrients such as protease inhibitors or saponins, which are known to interfere [38] with digestive enzyme secretion and activity. There was an expectation that the chitin present in MPWM might induce changes in the secretion of the analysed digestive enzymes, but that was not the case. Belghit et al. [2] also reported that insect meal had no effect on protease activity in Atlantic salmon, which was attributed to the addition of supplemental lysine and methionine during experimental diet formulation. The activity of intestinal chitinase was very low in the experimental dusky kob, which could be the reason for reduced feed intake at higher MPWM inclusion levels in the current study. However, the low chitinase activity did not negatively affect diet utilization, as seen in lower FCR and higher PER values as dietary MPWM levels increased. This provides evidence of the potential value of MPWM as an FM substitute in juvenile dusky kob diets. The observation that the carnivorous juvenile dusky kob had low chitinase activity is similar to a report by Kroeckel et al. [28] on the carnivorous turbot (*Psetta maxima*) reared on diets containing black soldier fly.

The effect of dietary chitin in insect meals on fish growth performance can be positive or negative depending on the fish species as well as the level of dietary chitin. In carp, tilapia, and Atlantic salmon, less than 1% dietary chitin reduced feed intake and ultimately growth performance [27,39,40]. On the other hand, fish with high chitinase activity, such as red sea bream (*Pagrus major*) (7400 $\mu$g NAG h$^{-1}$ g$^{-1}$), Japanese eel (*Anguilla japonica*) (3500 $\mu$g NAG h$^{-1}$ g$^{-1}$), yellowtail kingfish (*Seriola quinqueradiata*) (2700 $\mu$g NAG h$^{-1}$ g$^{-1}$), cobia (*Rachycentron canadum*) (3075 $\mu$g NAG h$^{-1}$ g$^{-1}$), and Atlantic cod (*Gadus morhua*) (500–1900 $\mu$g NAG h$^{-1}$ g$^{-1}$) showed no reduction in feed intake and growth with diets containing 10% chitin [30,31,41–43].

*4.2. Blood Parameters*

It is universally accepted that blood makes up between 1.3 and 7% of the total weight of fish and contributes immensely to metabolic processes of gas exchange between an organism and its environment. This makes blood parameters important indicators of the physiological condition or stress response to endogenous or exogenous stimuli–including diets [44]. It has been reported that changes in haematological parameters depend on several factors, which include fish age or weight at the time of blood sampling, and are not necessarily a negative response to what it has consumed [45,46]. There was generally no discernible pattern in haematological parameters in response to increment levels of MPWM in the diets, and the parameters were within the normal range for dusky kob, as previously reported in our earlier studies [3,18]. The current observation, which implies that MPWM did not infringe on the ability of dusky kob to fend off diseases, is in concordance with a list of studies with other insect meals in aquaculture fish that reported normal haematological parameters [1,47,48]. Serum biochemical indices from fish fed MPWM diets were similar to control-fed fish. These observations were consistent with those of Belghit et al. [2] when feeding Atlantic salmon with black soldier fly meal as well as those of Zhou et al. [1] who fed black soldier fly meal to carp. There were no dietary effects on the activities of serum aspartate aminotransferase or alanine aminotransferase in the current study, suggesting that MPWM does not impair liver function [49]. Similar blood total protein, albumin, and globulin levels across the four dietary groups further suggests that the nutritional status of the dusky kob was not compromised by replacing FM with MPWM.

**5. Conclusions**

The search for affordable and locally available protein sources that can be used in place of FM is critical for sustainable marine fish aquaculture in southern Africa. Our pilot

study provides evidence that while MPWM levels beyond 11.13% compromised overall feed intake, no changes were observed in feed utilization efficiency, blood parameters, and growth performance of juvenile dusky kob. Critically, FCR and PER improved as the level of dietary MPWM increased. The study further revealed that the MPWM does not interfere with intestinal amylase, lipase, and protease activities. No chitinase activity was detected in the experimental fish across all diets. It was concluded that up to 11.13% MPWM can be used in place of FM in commercial juvenile dusky kob diets.

**Author Contributions:** T.C.N., V.M., M.J.M., M.M. and O.C.W. were equally involved in the conceptualization and formulation of the research question and execution of the feeding trial, data analysis, and manuscript writing and editing. All authors have read and agreed to the published version of the manuscript.

**Funding:** This research received no external funding.

**Institutional Review Board Statement:** The study was conducted according to the guidelines of the Declaration of Helsinki and approved by the Animal Research Ethics Committee of the University of Mpumalanga (FANS17; 21 May 2020).

**Informed Consent Statement:** Not applicable.

**Data Availability Statement:** The data presented in this study are available on request from the corresponding author. The data are not publicly available because the diet formulae used in this study are proprietary brands.

**Acknowledgments:** The University of Mpumalanga is hereby acknowledged for providing funding for the production of the experimental diets. We are immensely grateful to the Department of Forestry, Fisheries and the Environment for giving us access to their facility and personnel in Cape Town to conduct the feeding trial. Freddy Manyeula from Botswana University of Agriculture and Natural Resources is acknowledged for assisting in the sourcing of the mopane worms.

**Conflicts of Interest:** The authors declare no conflict of interest.

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
