# Peer review of "Partial Substitution of Fishmeal with Mopane Worm Meal in Dusky Kob Fingerling (Argyrosomus japonicus) Diets: Feed Utilization, Digestive Enzyme Activity, Blood Parameters, and Growth Performance"

_2673-9496, doi:10.3390/aquacj2020006_

Round 1
Reviewer 1 Report
Topic of the study has still actuality due to decreasing availability and increasing price of fish meal and increasing demand for novel feed ingredients.
The study is focusing to find the out the optimal inclusion level of mopane worm in the diets of dusky kob fingelings. The aim of the work is well defined, composition and structure of the manuscript is logic, well-written. However, there are different issues in the paper that needs further major revision. Moreover, there is a core problem in the formulation of the diets and their nutritional value. The diets are not isonitrogenous, isolipidic and isoenergetics. Insect meal was included into diet not only for replace fish meal thus all ingredients. This fact rises several questions: which effect could be detected now, increased poultry or insect meal level or decreased fish meal level?
As conclusion, all the tested feeds should be considered as suitable diets for fish. Comparison of the diets with different inclusion level and defining an optimum level has not sense in this case.
There are still some mistakes observed in the manuscript:
LINE 151 weekly measurements of body mass of individual fish (100 pc) or tank fish biomass?
LINE 153 CALCULATIONS
Is missing the formulas for FI= feed intake (feed offered or feed consumed g/fish)
Protein intake= protein (g)/fish ???
Is missing the exact information on the daily feed ratio. Was adjusted the daily feed according to the growth of fish based on the weekly measurements?
LINE 223 Table 1. in the last row of the Table 1 there is wwwwssswornm, please clear it.
Is missing the origin or the distributor name of each ingredients, and where is relevant the protein content: e.g. fishmeal 60 or poultry meal 65. Wheat gluten or corn gluten was used?
Proximate composition is: dry weight basis or original weight (as is)?
LINE 283 Table 4 and 5: 2 SEM and 3 Significance. Please correct it.
Creatinine level in MPWM6 treatment is much higher compared to other diets. Surely are not significant differences between the treatments?
LINE 300 Chitinase activity is not higher in MPWM18 group as was mentioned in LINE 296. (see Table 6).
There is not presented SEM data, according to this clear this footnote and indicate the standard deviation or SEM in that table
LINE 333 Please correct it: “Feed conversion ratio in fish fed MPWM-containing diets was lower than that of the control diet, possibly a reflection of reduced feed intake in MPWM diets compared to the control”. The feed intake is not reduced in the MPWM diets compared to control! Please see Table 3.
LINE 414 Conclusion: The feed intake data are not relevant to calculate the optimum inclusion level of the insect. The MPWM 6 diet has higher fat, MPWM3 has high protein content compared to control and MPWM18.
The level of 11.13% inclusion is a wrong conclusion.

Reviewer 2 Report
The MS is interesting and reports a new cheaper protein source for fish feed. It is clear, with specific objectives, adequate methodology in some items, but in some points it needs to be better detailed (see below). The results are clear and the discussion can be improved, reduced and more direct. Some points need to be changed:
- Abstract: put the meaning of the acronym MPWM, as it is the first time it appears.
- Introduction: line 46 the scientific name Imbrasia Belina should be changed to Imbrasia belina
- throughout the text, correct mg/L to mg L-1. This is the correct nomenclature for all other units.
- transforms rpm of hematocrit analysis to g
- the methodology of hematological (morphological) analyses, such as white blood cell counts, is incorrect. The results are in %, which doesn't mean much. It is needed to count the RBC first and the total number of leukocytes in smears, counting 4000 cells in each fish smear separating the White ones. The percentagem of each leukocyte is calculated in smears also, counting 400 white cells. With the percentage calculate, the total number of each leukocyte is knwon. This is a common mistake and should be avoided. Today we have precise methodologies for this, which do not require any special equipment, just a microscope. In addition, thrombocytes were counted along with leukocytes. These cells are another group, not belonging to white blood cells. The hematology discussion is flawed.
- Lines 220, 221 and 222: “The energy content of the four diets was similar, except for the MPWM18, which had numerically lower energy content than the rest of the diets.” Wouldn't it be the opposite, that is, is this the treatment with the highest energy content?
- the conclusions are almost all results and only the last sentence is really a conclusion.
Refrences: The following references were not found in the text: Daniel (2004), Folch et al (1957), Hepher (1988), Hua et al. (2019), Lock et al. (2016), Makhado et al. (2015) and Soetemans et al. (2020). The reference Zhou et al. (2016) – (line 407) is in the text but not in the listing.
Reviewer 3 Report
Review for the paper "Partial substitution of fishmeal with mopane worm meal in dusky kob fingerling (Argyrosomus japonicus) diets: Feed utilization, digestive enzyme activity, blood parameters, and growth performance" by Tshegofatso C. Nyuliwe, Victor Mlambo, Molatelo J. Madibana, Mulunda Mwanza and Obiro C. Wokadala submitted to "Aquaculture Journal".
General comment.
Development of sustainable and low-cost fish aquaculture requires new low-cost aquafeeds providing suitable sources of protein, fat and other essential substances. Fishmeal widely used in aquaculture is considered a relatively high-cost component of fish diets and, therefore, there is a growing demand for alternative sources of high-quality protein for fish farming. Different species of insects, or more precisely, their larval stages have been tested as a replacement for fishmeal over the past decades worldwide. In South Africa, the mopane worm is an example of such an organism that was included in the diet of farmed fish. Previous studies have been focused on freshwater fish species and demonstrated contradictory results. Taking into account the knowledge gap on the effects of mopane worm as a food source on marine fish species, the authors conducted a classical experimental study to reveal the possibility to replace fishmeal with mopane worm in the diet of dusky kob fingerlings, a promising fish species for the local industry. The authors found that the inclusion of mopane worm meal influenced the overall feed intake following the quadratic function. Overall weight gain and specific growth rate were found to linearly increase and overall feed conversion ratio to linearly decrease following an increase in mopane worm meal inclusion levels. Substituting fishmeal with mopane worm meal had no effects on digestive enzyme activities and hematological parameters. Among serum biochemical components, only urea demonstrated an increase as a response to higher mopane worm meal inclusion levels. The authors concluded that a replacing level of 11.13% of fishmeal in commercial dusky kob diets is optimal as it does not compromise feed intake, feed utilization efficiency, growth rate, and physiological status of juvenile dusky kob. Standard rearing methods and methods to collect samples and to treat the data were used in the study. Experiments were performed according to international guides and standards. Main results are illustrated with relevant Figures and Tables. Discussion is focused on the main findings. Statistical methods are adequate and correctly used. I have two suggestions to improve the ms prior to the ms being accepted for publication in "Aquaculture Journal".
General recommendations.
1) The authors should format citations and references according to the MDPI standards.
2) Please, check the header of the last column in Table 1. Please, check the position of "5.36" (Table 1, column 2, line "Fish and poultry oil"
Specific comments
L 27. Consider replacing " haematology" with " haematological".
L 52. Consider replacing " most of rural" with " most rural".
L 58. Consider replacing "source" with "source of".
L 74. Consider replacing "effect on" with "effect of".
L 145. Consider replacing "observed with" with "observed in".
L 176. Consider replacing "Biuret method" with "the Biuret method".
L 204. Consider replacing "feed conversion ration" with "feed conversion ratio".
L 212. Consider replacing "significance" with "the significance".
L 221. Consider replacing " which had" with " which had a".
L 228. Consider replacing " specific" with "for specific".
L 283. Consider replacing " haematology" with " haematological".
L 336. Consider replacing "contrast with" with "contrast to".
L 341. Consider replacing " purpose" with " purposes".
L 348. Consider replacing " suggests" with " suggest".
L 362. Consider replacing " activity on" with " activity of".
L 381. Consider replacing "level" with "the level".
L 398. Consider replacing " haematology" with " haematological".
L 400. Consider replacing " within normal range" with " within the normal range".
L 402. Consider replacing " fend of" with " fend off".
L 404. Consider replacing " haematology paramters" with " haematological parameters".
L 416. Consider replacing "This pilot" with "Our pilot".
L 459-460. The authors should fix this error: one reference is formatted as two.
References: Some Latin names are presented without italics.
Round 2
Reviewer 1 Report
Accept the utilization of the calculated feed data (using the calculation of the feed formulation software) instead of the measured in laboratory condition. In this case you have to modify in the Chapter 2.4 description of the analysis of the diet according as you did.
It has been changed the gross energy to digestible energy (DE) and metabolizable energy (ME). In the Chapter 2.4 description of measurements of the gross energy is given (LINE 117). Please clarify which one would you like to use (or both three), but give the information on calculation for DE and ME.
